# Evaluating the Abilities of Satellite-Derived Burned Area Products to Detect Forest Burning in China

Xueyan Wang [1], Zhenhua Di [1,2,*] and Jianguo Liu [2]

1 State Key Laboratory of Earth Surface Processes and Resource Ecology, Faculty of Geographical Science, Beijing Normal University, Beijing 100875, China; 202121051041@mail.bnu.edu.cn
2 Key Laboratory of Intelligent Control Technology for Wuling-Mountain Ecological Agriculture in Hunan Province, School of Mathematics and Computational Science, Huaihua University, Huaihua 418008, China; jgliu@mail.iap.ac.cn
* Correspondence: zhdi@bnu.edu.cn; Tel.: +86-10-5880-4622

**Abstract:** Fire plays a prominent role in the construction and destruction of ecosystems, and the accurate estimation of the burned area (BA) after a fire occurrence is of great significance to protect ecosystems and save people's lives and property. This study evaluated the performances of three publicly available BA satellite products (GFED4, MCD64CMQ, and FireCCI5.1) in detecting Chinese forest fire burning from 2001 to 2016 across different time scales (yearly, monthly, and seasonally) and spatial scales (regional and provincial). The reference data were derived from the monthly China Forestry Statistical Yearbook (CFSY), and they were mainly used to evaluate the detection ability of each of the three BA products in the three major forest fire areas of China consisting of the Northeast (NE), Southwest (SW), and Southeast (SE) regions. The main results are as follows: (1) A significant declining BA trend was demonstrated in the whole study area and in the NE and SE subregions. Specifically, the slopes for the whole area ranged from −3821.1 ha/year for MCD64CMQ to −33,218 ha/year for the CFSY, the slopes for the NE region ranged from −3821.1 ha/year for MCD64CMQ to −33,218 ha/year for the CFSY, and the slopes for the SE region ranged from −594.24 ha/year for GFED4 to −3162.1 ha/year for the CFSY. The BA in China was mainly dominated by forest fires in the NE region, especially in 2003 and 2006 when this region accounted for 90% and 87% of occurrences, respectively. (2) Compared with the CFSY, GFED4 had the best performance at the yearly scale with an RMSE of $23.9 \times 10^4$ ha/year and CC of 0.83. Similarly, at the monthly scale, GFED4 also had the best performance for the three regions, with the lowest RMSE ranging from $0.33 \times 10^4$ to $5.4 \times 10^4$ ha/month—far lower than that of FireCC5.1 which ranged from $1.16 \times 10^4$ to $8.56 \times 10^4$ ha/month (except for the SE region where it was slightly worse than MCD64CMQ). At the seasonal scale, GFFD4 had the best performance in spring and winter. It was also noted that the fewer BAs in summer made the differences among the products insignificant. (3) Spatially, GFED4 had the best performance in RMSEs for all the provinces of the three regions, in CCs for the provinces of the SW and SE regions, and in MEs for the provinces of the SE region. (4) All three products had stronger detection abilities for severe and disaster fires than for common fires. Additionally, GFED4 had a more consistent number of months with the CFSY than the other products in the NE region. Moreover, the conclusion that GFED4 had the best performance in the China region was also proved using other validated BA datasets. These results will help us to understand the BA detection abilities of the satellite products in China and promote the further development of multi-source satellite fire data fusion.

**Keywords:** forest fires; burned area; evaluation of BA satellite products



## 1. Introduction

Forest fires constitute an increasingly important ecological disturbance factor in achieving the sustainable management of forest resources, particularly with the intensification of

climate change and human activity [1]. The burned area (BA) is one of the most important parameters in fire management. It not only provides critical information about fire regimes, but also assists in calculating carbon, trace gas, and aerosol emissions [2]. Therefore, accurately quantifying BAs is essential for understanding the effects of biomass burning on the carbon–nitrogen cycle at a global or regional scale [3].

There are four common types of forest fire detection methods [4]: (1) ground patrol [5], (2) near-surface detection [6], (3) aircraft patrol [7], and (4) satellite remote-sensing detection. Initially, a number of coarse- and medium-resolution sensors were used to detect active fires, such as the Advanced Very High Resolution Radiometer (AVHRR) [8], the Visible and Infrared Scanner (VIRS) [9], and the Moderate Resolution Imaging Spectro-radiometer (MODIS) [10]. However, the active fire products only detected the locations and timings of burning fires when their satellites passed over. Obviously, the detected BA data was discontinuous and unreliable. Then, the BA mapping method was developed. For instance, Kasischeke and French [11] generated a BA map for Alaskan boreal forest fires during 1990 to 1991 by applying the differencing method to 15-day AVHRR Normalized Difference Vegetation Index (NDVI) data, and Barbosa et al. [12] mapped BAs in Africa using daily 5 km AVHRR imagery and information on changes in reflectance, brightness, temperature, and a vegetation index. However, these methods do not exploit active fire information and thus create inaccuracies in the BA products. Hybrid algorithms can supplement the drawbacks of previous remote sensing methods, combining the advantages of remotely sensed indicators (e.g., surface reflectance, surface temperature, NDVI) and active fire maps. Giglio et al. [13] presented an automated method of BA mapping by combining 500 m MODIS imagery and 1 km MODIS active fire observation BA data. Zhang et al. [14] constructed a fire label dataset using the VNP14IMG fire product and Himawari-8 multiband data that includes active fires. Additionally, the products like GFED4, MCD64CMQ, and FireCCI5.1, which were used in this study, were developed using the hybrid algorithm.

Obviously, uncertainties exist among fire satellite products due to differences in their sensors or inversion algorithms. Some studies have evaluated the performances of several BA products on a global scale [15–18]. In addition, more studies have focused on the assessments of different climate subregions. For instance, Núñez-Casillas et al. [19] and Ruiz et al. [20] evaluated the performance of different BA products in the Canadian boreal forest region. Other studies focused on the validation and comparisons of BA products for the tropical regions of South America and Africa [21–23]; evaluations were also conducted in other areas such as Northeastern Asia [24] and Mediterranean eco-systems [25]. There are also some studies on the evaluation of BA products in China. Chen et al. [26] evaluated the performance of three common satellite BA products in eastern China from 2018 to 2019, Jiao et al. [27] evaluated the performance of four BA products in the fire-prone, cloudy, and mountainous areas of the Sichuan province from 2013 to 2018, and Zhang et al. [28] evaluated the abilities of three BA products to estimate the total BA for cropland regions in China from 2005 to 2009. There appear to be few studies on the comprehensive evaluation of satellite fire BA products for forest regions in China. Moreover, the evaluation periods in the existing studies were relatively short, usually comprising only three to six years.

Therefore, this study evaluates the performances of three popular satellite BA products on the three large forest fire-prone regions of China, covering a relatively long period from 2001 to 2016. The rest of the paper is organized as follows. Section 2 introduces the study area, fire BA data, and evaluated metrics. The evaluation results at the temporal and spatial scales are demonstrated in Section 3. The comparison of the number of months for the different BA classes is also included in this section. The discussion and conclusions are presented in Sections 4 and 5, respectively.

## 2. Materials and Methods

### 2.1. Study Area

The study area (Figure 1) consists of three regions of China: the Northeast (NE) region, Southwest (SW) region, and Southeast (SE) region. Based on the forest area information

from the China Environment Statistical Yearbook, the NE, SW, and SE are the three major regions with the highest forest area in China. Among them, the Inner Mongolia and Heilongjiang provinces in the NE region and the Yunnan province in the SW region have the largest forest area among the 16 provinces of three regions. In total, the forest area of the three regions accounts for 85.14% of the total forest area in China. Notably, 95.66% of the country's forest fire area consisted of these three regions from 2001 to 2016.

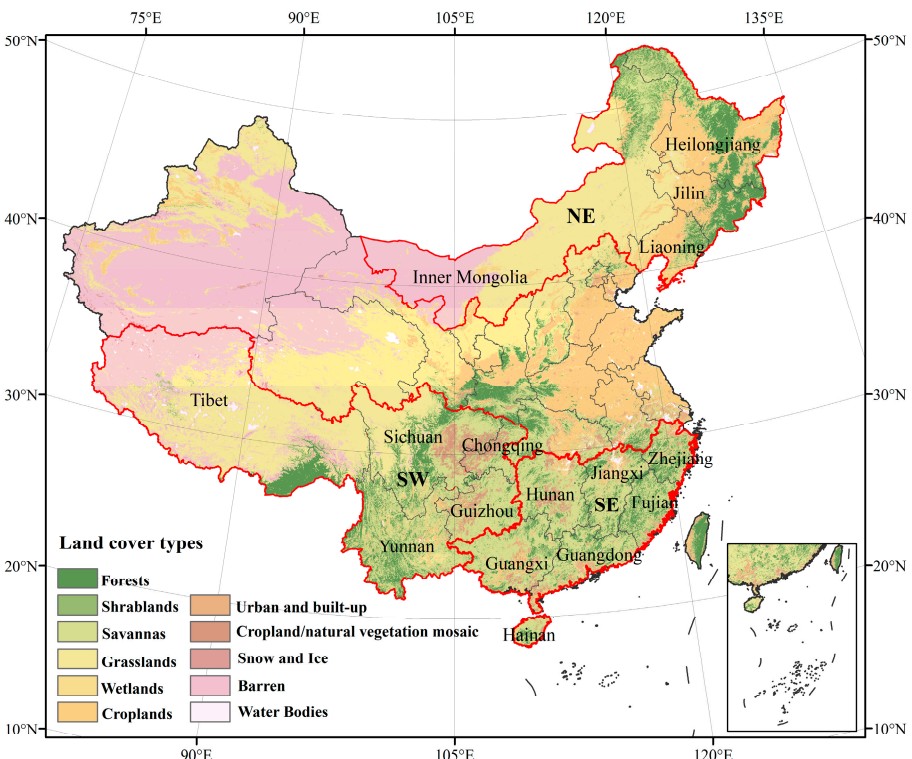

**Figure 1.** Land cover map of China in 2010 with the three study areas including the Northeast (NE) region, Southwest (SW) region, and Southeast (SE) region.

The three regions have significant differences in climate and vegetation. Specifically, the NE region is characterized by higher windspeed, colder climate, and heavier snowfall, but less rainfall, compared to Southern China. There are many mountains with altitudes of 300 to 1400 m in the north of NE, primarily covered by deciduous and coniferous forests in cold temperature zones. The eastern plains with an altitude below 200 m are densely populated and primarily covered by the deciduous mixed broadleaf–conifer forests in the temperate zone. The NE region has extensive forests that are rarely inhabited by humans, and here, forest fires are characterized by rapid spread and extensive burned area; the SW region is characterized by cloudy and foggy weather in spring and autumn, with annual precipitation ranging from 600 to 2300 mm. The elevation of most areas is between 500 and 6000 m, and the vegetation displays a distinct vertical distribution pattern from subtropical broad-leaved evergreen forest to alpine grassland. Furthermore, forests are primarily distributed in high mountain valleys or dry river valleys, which results in the unpredictability and uncontrollability of forest fires due to the complex terrain and climatic background of the region; the SE region mainly consists of hills below 500 m, characterized by large amounts and high intensity of rainfall over 800 mm. It is mainly covered by subtropical broad-leaved evergreen forest, and the high density of forest coverage makes it an important production base for Chinese-specialty forest products.

*2.2. Forest Fire BA Datasets*

The forest fire BA datasets used in this study include one official statistical dataset for validation and three different satellite products for evaluation. The evaluation period

lasted from 2001 to 2016. The spatial resolution of the official statistics is given by province and that of the satellite products is a uniform 0.25°. Their unified time resolution is one month. Note that the BA fractions in each cell for satellite products are provided for the different land cover types; therefore, the forest fire areas were quantified by overlapping the forest coverage and the satellite fire BA distribution.

### 2.2.1. Official Statistics

The China Forestry Statistical Yearbook (CFSY), published by the National Forestry and Grassland Administration of China (NFGA), provides information on forest fires every year. It refers to the number of forest fires, the burned area (BA), and the corresponding economic losses. The forest fire dataset in the CFSY (www.chinayearbooks.com, accessed on 1 May 2022) was collected from the NFGA's subordinate administrative departments such as the provincial, municipal, and county forestry bureaus. As a government agency, one of the duties of the NFGA is to record the forest fire information like BA by employing specialized officers; therefore, the accuracy of the data issued by them is high. The collected BA has a spatial resolution based on province and a temporal resolution based on month. In this study, the provincial BA data at monthly scale from the CFSY were used as a reference dataset to evaluate the results from the satellite-derived BA products for the three major forest areas in China.

### 2.2.2. GFED4

The Global Fire Emissions Database (GFED4, available from sftp://fuoco.geog.umd.edu/data/GFED/GEFD4, accessed on 1 May 2022) provides global monthly BA data from June 1995 to 2016 with a spatial resolution of 0.25 degrees. The data were derived by the 500 m Collection 5.1 MCD64A1 product. Giglio et al. [13] applied the MODIS DB burned-area mapping algorithm on this dataset to obtain BA maps, extended the time series back to 1997, and combined it with active fire data from the Tropical Rainfall Measuring Mission (TRMM) Visible and Infrared Scanner (VIRS) and the Along-Track Scanning Radiometer (ATSR) sensors. We used 2 of the 7 data layers in the product, namely, Burned Area and Land Cover Dist. The dataset is widely used for large-scale modeling studies.

### 2.2.3. MCD64CMQ

MODIS C6 provides global BA data from November 2000 to September 2022 with a spatial resolution of 0.25° (sftp://fuoco.geog.umd.edu/data/MODIS/C6/MCD64CMQ, accessed on 1 May 2022). The MCD64CMQ dataset was generated from the MODIS C6 Terra and Aqua 500 m daily surface reflectance products (MOD09GHK and MYD09GHK), 1 km daily level 3 MODIS active fire products (MOD14A1 and MYD14A1), and the MCD63A1 algorithm-based MCD12Q1 500 m annual land cover product [29]. Additionally, the BA in C6 is more sensitive to small and moderate burns, which reduces the data uncertainty compared to C5.1. The MCD64CMQ product includes 4 data layers, namely, Burned Area, QA (8-bit quality assurance bit field), Unmapped Fraction, and Land Cover Dist. Finally, we obtained the forest BA fraction from the 17 land cover types archived in the MCD12Q1 product.

### 2.2.4. FireCCI5.1

The latest FireCCI5.1 version, funded by the European Space Agency's Climate Change Initiative project, provides global BA data from 2001 to 2019 (https://cds.climate.copernicus.eu/cdsapp#!/dataset/10.24381/cds.f333cf85?tab=form, accessed on 1 May 2022). The product algorithm integrates the MOD09GQ 250 m daily reflectance measurements and MCD14ML 1 km daily active fire data to determine the BA map through a two-phase approach (seed detection and region growing). It is an improvement when compared with the previous FireCCI5.0 and FireCCI4.1 versions. The product includes 3 data layers, namely, JD—date of first detection, CL—confidence level, and LC—the type

of land cover when a pixel burns. Therefore, the CCI-LC was used to indicate the forest fire BA.

In summary, the MODIS sensor on the Terra and Aqua platforms is one of the important detection tools for global BA data. The MCD64 and FireCCI51 products are often compared in pairs and have provided BA data in numerous fire assessment studies. Similarly, GFED4 is widely used to help large-scale atmospheric and biogeochemical models explain the impacts of climate change and land management on fire activities. China has vast territory and diverse climate, and there have been few studies on the comprehensive evaluation of satellite BA products in China rather than in its specific regions. Therefore, considering the heterogeneity characteristics of China, these three BA products with long time series were selected to evaluate their detection abilities for forest fire BAs in China.

### 2.3. Methods

#### 2.3.1. Preprocessing of BA Satellite Products

The HDF files of GFED4 and MCD64CMQ and NC files of FireCCI5.1were converted into GeoTIFF format to demonstrate the BA spatial distribution. Then, the BA data were extracted from the GeoTIFF using the MATLAB software with R2020b v9.9.0 to conduct a comparison with CFSY. The data types of the products used in this study were grid versions with $0.25°$ spatial resolutions.

#### 2.3.2. Validation Metrics

To evaluate the performances of BA grid products, a comparison of three products with CFSY was conducted at three scales (national, regional, and provincial). Three commonly used metrics, the correlation coefficient (CC), root-mean-square error (RMSE), and mean error (ME), were adopted for the assessment, and their expressions are shown in Table 1. Note that there are some commonly used BA evaluation metrics such as the commission error ratio, the omission error ratio, and the Dice coefficient, which combines the two previous metrics [15,19,20,25]. However, limited by the scale of the referenced dataset, they were not used instead of RMSE, ME, and CC in this study.

**Table 1.** Statistical metrics used in this study.

| Statistics | Formula * | Perfect Value |
|---|---|---|
| Correlation coefficient (CC) | $CC = \dfrac{\Sigma\left(G_i - \bar{G}\right)\left(O_i - \bar{O}\right)}{\sqrt{\Sigma\left(G_i - \bar{G}\right)^2 \Sigma\left(O_i - \bar{O}\right)^2}}$ | 1 |
| Root-mean-square error (RMSE) | $RMSE = \sqrt{\dfrac{1}{n}\sum_{i=1}^{n}(G_i - O_i)^2}$ | 0 |
| Mean error (ME) | $ME = \dfrac{1}{n}\sum_{i=1}^{n}(G_i - O_i)$ | 0 |

* $G_i$ and $O_i$ represents the satellite-derived BA product and BA dataset of CFSY, respectively.

## 3. Results

### 3.1. Comparisons of Annual BA

Figure 2 shows the spatial distributions of the total BA registered in the three products for the Chinese mainland from 2001 to 2016. It was obvious that the fire BA was mainly concentrated in the NE, SW, and SE regions in China. Specifically, it presented in the north of the NE region, the south of the SW region, and the central part of the SE region. Overall, GFED4 detected the least fire BA and FireCCI5.1 detected the most fire BA. Table 2 lists the total BA value for the Chinese mainland and its three main fire regions (i.e., NE, SW, and SE). The ratio of FireCCI5.1 to GFED4 varied from 8.57 for the SE region to 3.97 for the SW region.

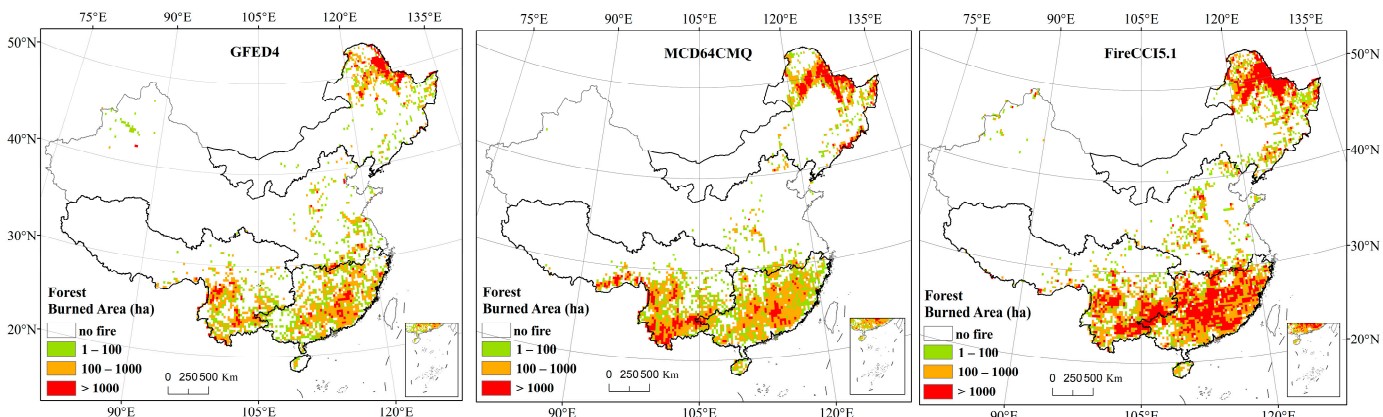

**Figure 2.** Spatial patterns of the total BA for the three satellite products in the Chinese mainland.

**Table 2.** Summaries of the three-gridded satellite burned area (BA) products for the Chinese mainland and its three main fire regions (Northeast (NE), Southwest (SW), and Southeast (SE)) from 2001 to 2016. (Unit: ha).

| Dataset | Northeast | Southwest | Southeast | Chinese Mainland |
|---------|-----------|-----------|-----------|------------------|
| GFED4 | 914,257 | 262,617 | 277,722 | 1,631,890 |
| MCD64CMQ | 1,178,805 | 631,376 | 347,604 | 2,217,534 |
| FireCCI5.1 | 5,162,262 | 1,042,291 | 2,381,047 | 8,917,285 |

Figure 3 shows the annual variations in the total BA from the four datasets for the Chinese mainland and the three regions from 2001 to 2016. A declining BA trend was demonstrated in the Chinese mainland and in the NE and SE regions, while the SW region showed an insignificantly declining trend. Specifically, the slopes for the whole area ranged from −3821.1 ha/year for MCD64CMQ to −33,218 ha/year for the CFSY, the slopes for the NE region ranged from −3821.1 ha/year for MCD64CMQ to −33,218 ha/year for CFSY, and the slopes for the SE region ranged from −594.24 ha/year for GFED4 to −3162.1 ha/year for the CFSY. Based on the y-axis information in Figure 3, it was found that the BA in China was mainly dominated by forest fires in the NE region. For instance, the CFSY in the Chinese mainland showed two unusually high values in 2003 and 2006, with the NE region accounting for 90% and 87%, respectively. Compared with the CFSY, GFED4 was the best-performing product at the Chinese mainland and its subregions except for MCD64CMQ in the SE. Although MCD64CMQ slightly outperformed GFED4 in the SE region, the total BA amount in the SE was far lower than that in the other regions (e.g., NE); therefore, GFED4 had the best overall performance in the Chinese mainland.

The scatter plots of the annual BA for the satellite products versus the CFSY are shown in Figure 4. For the Chinese mainland, the annual RMSE ranged from $23.9029 \times 10^4$ to $41.4291 \times 10^4$ ha/year and the annual CC ranged from 0.4778 to 0.8304. Regionally, the ranges of the annual RMSEs were from $20.3887 \times 10^4$ to $28.5668 \times 10^4$ ha/year in the NE region, $1.7143 \times 10^4$ to $6.4769 \times 10^4$ ha/year in the SW region, and $2.8129 \times 10^4$ to $12.9392 \times 10^4$ ha/year in the SE region. The ranges of annual CCs were from 0.47 to 0.9184, 0.2790 to 0.6597, and 0.5272 to 0.68, respectively. Among them, the GFED4 estimates showed the best performance in all regions, with the lowest RMSE (except for the SE, where MCD64CMQ performed slightly better). In contrast, FireCCI5.1 had the worst RMSE and worst ME for all four regions. Notably, MCD64CMQ had the smallest CCs (except in the SE region, where it had the highest CC value—0.68).

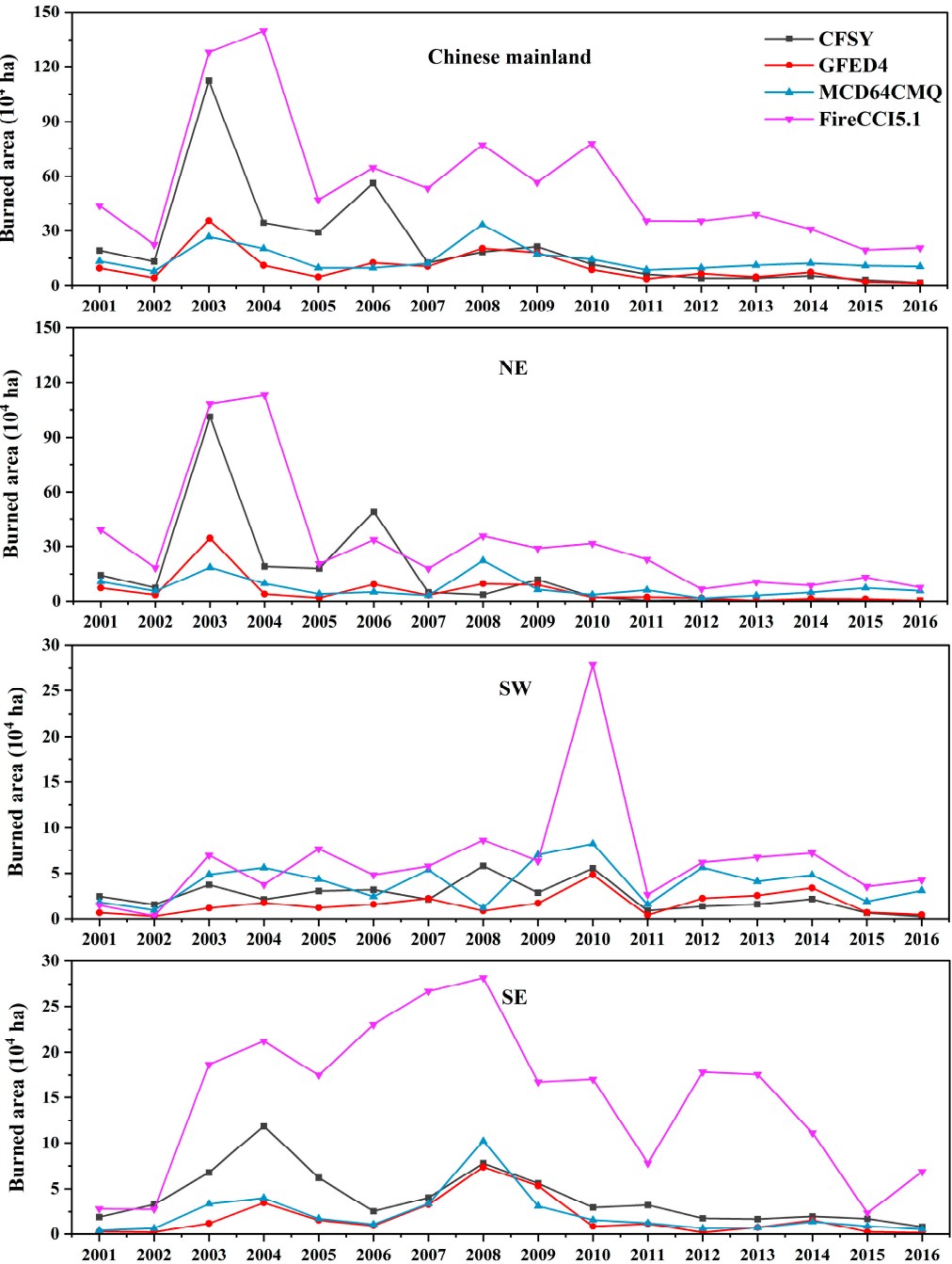

**Figure 3.** The annual changes in total BA in the Chinese mainland and the three subregions from 2001 to 2016.

### 3.2. Comparison of Monthly and Seasonal BA

Figure 5 shows the variations in average monthly BA values for the four datasets in the Chinese mainland and three regions. Apparently, all products captured significant seasonal variations. Forest fires in the NE region mainly occurred in spring (MAM) and autumn (SON), while in southern China (SW and SE regions) they occurred mainly in spring (MAM) and winter (DJF). Moreover, the SE region in autumn was more prone to forest fires than the SW region. The results demonstrated three major aspects. First, the accuracies of the BA estimates decreased as the BA increased; second, the estimates of FireCCI5.1 were significantly high in some individual months, indicating the importance of evaluating the product's detection capability during the fire season; third, the overall performance of GFED4 was better than that of MCD64CMQ, except for some months or

seasons when MCD64CMQ slightly outperformed GFED4, such as April in the SW region and autumn (SON) in the NE region and Chinese mainland. In other words, GFFD4 had the best performance in spring (MAM) and winter (DJF). For summer, there was a smaller BA; therefore, the differences between the three products were small. Overall, GFED4 performed the best at the monthly and seasonal scales.

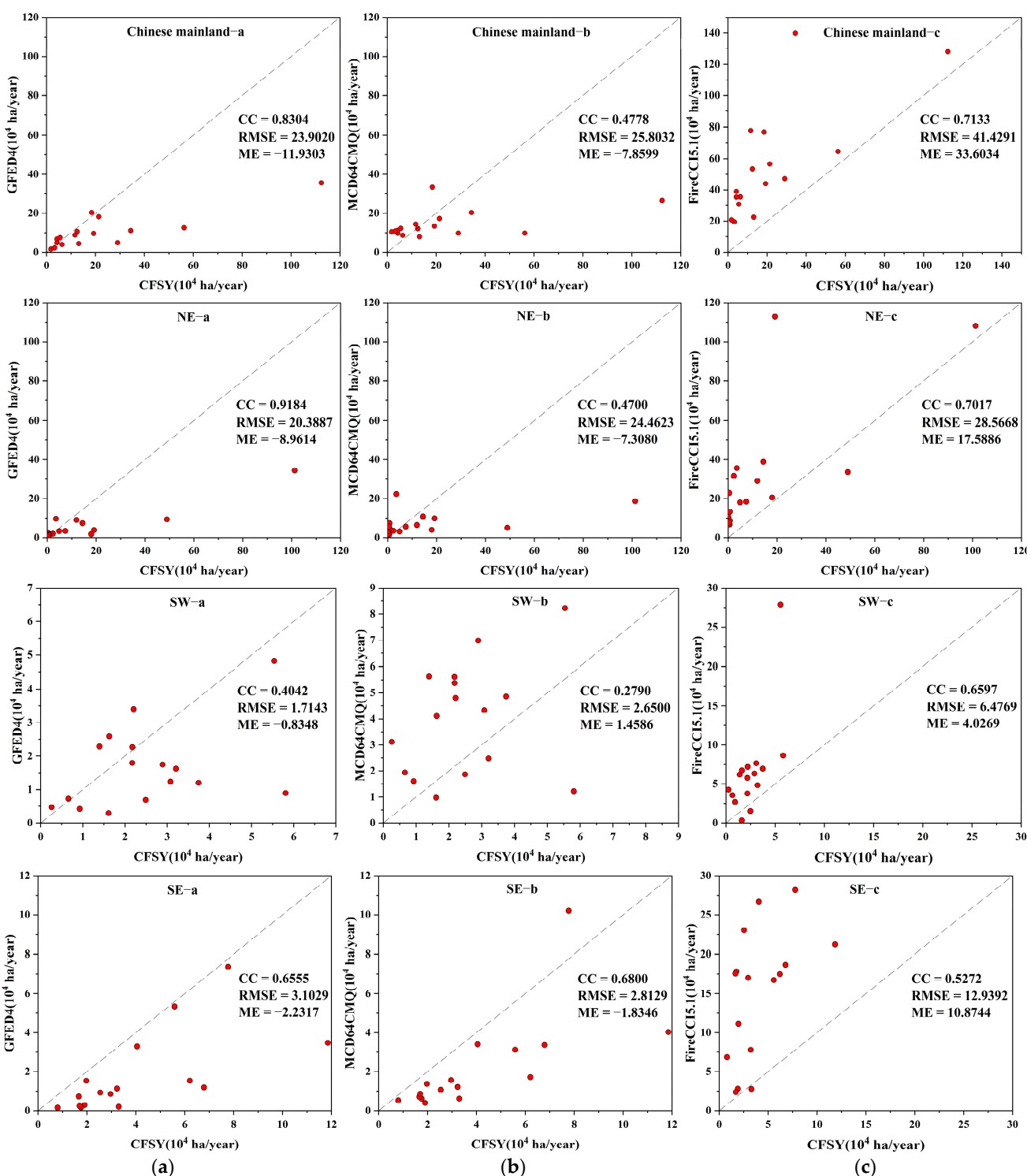

**Figure 4.** Scatter plots of annual BA in the Chinese mainland and the three subregions for the CFSY versus the three satellite estimates: (**a**) GFED4, (**b**) MCD64CMQ, and (**c**) FireCCI5.1.

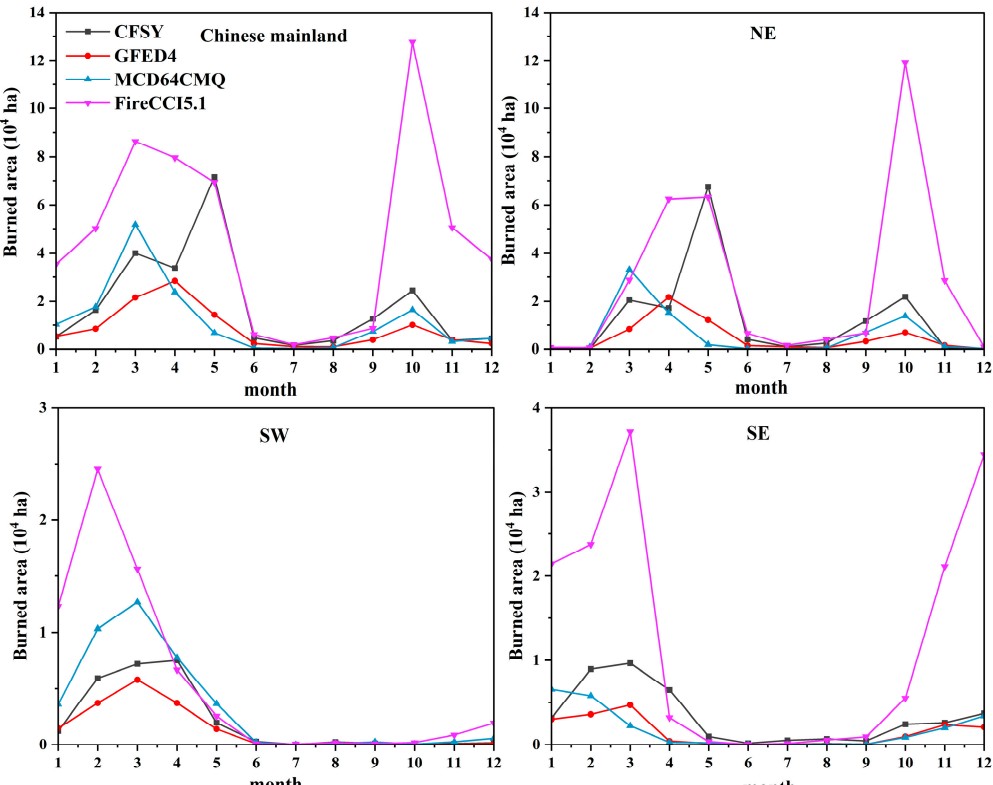

**Figure 5.** Mean monthly BA from 2001 to 2016 for the four datasets in the Chinese mainland and the three subregions.

The scatter plots of monthly BA for the satellite products versus the CFSY are shown in Figure 6. For the Chinese mainland, the CCs for GFED4, MCD64CMQ, and FireCCI5.1 were 0.5815, 0.3618, and 0.5564, respectively, their RMSE values were $5.4227 \times 10^4$, $5.8516 \times 10^4$, and $8.5644 \times 10^4$ ha/month, respectively, and their ME values were $0.9942 \times 10^4$, $0.6550 \times 10^4$, and $-2.8003 \times 10^4$ ha/month, respectively. For the three regions, the monthly RMSE for the NE, SW, and SE regions ranged from $5.2568 \times 10^4$ to $7.6352 \times 10^4$ ha/month, $0.3256 \times 10^4$ to $1.1608 \times 10^4$ ha/month, and $0.4793 \times 10^4$ to $2.5655 \times 10^4$ ha/month, respectively. The monthly CC for the NE, SW, and SE regions ranged from 0.2935 to 0.6178, 0.6384 to 0.7197, and 0.4925 to 0.7622, respectively. The results showed that the best performance occurred in the GFED4 estimate, which provided the lowest RMSE and highest CC in the Chinese mainland and the three subregions. Therefore, it can be concluded that the GFED4 estimate was optimal among the three satellite products at the monthly and annual scales.

The specific seasonal differences between the three products and CFSY are shown in Table 3. In spring, there were the largest amounts of BA (as shown in Figure 5), and GFED4 had the best performance according to the statistic indices except for RMSE in the NE region and CC in the SW region. For summer, which had the lowest amounts of BA, GFED4 also had the best performance in the NE and SW regions. The relatively severe fires mainly occurred in autumn; however, GFED4 had the second-highest performance following MCD64CMQ. There were relatively small amounts of BA in winter, when GFED4 again obtained the best performance, except for the CCs in the NE and SW regions. Overall, GFED4 had the best performance for the three regions in spring, summer, and winter, except for autumn when its performance was slightly worse than that of MCD64CMQ but still higher than that of FireCCI5.1. Overall, GFED4 had the best performance on the seasonal scale.

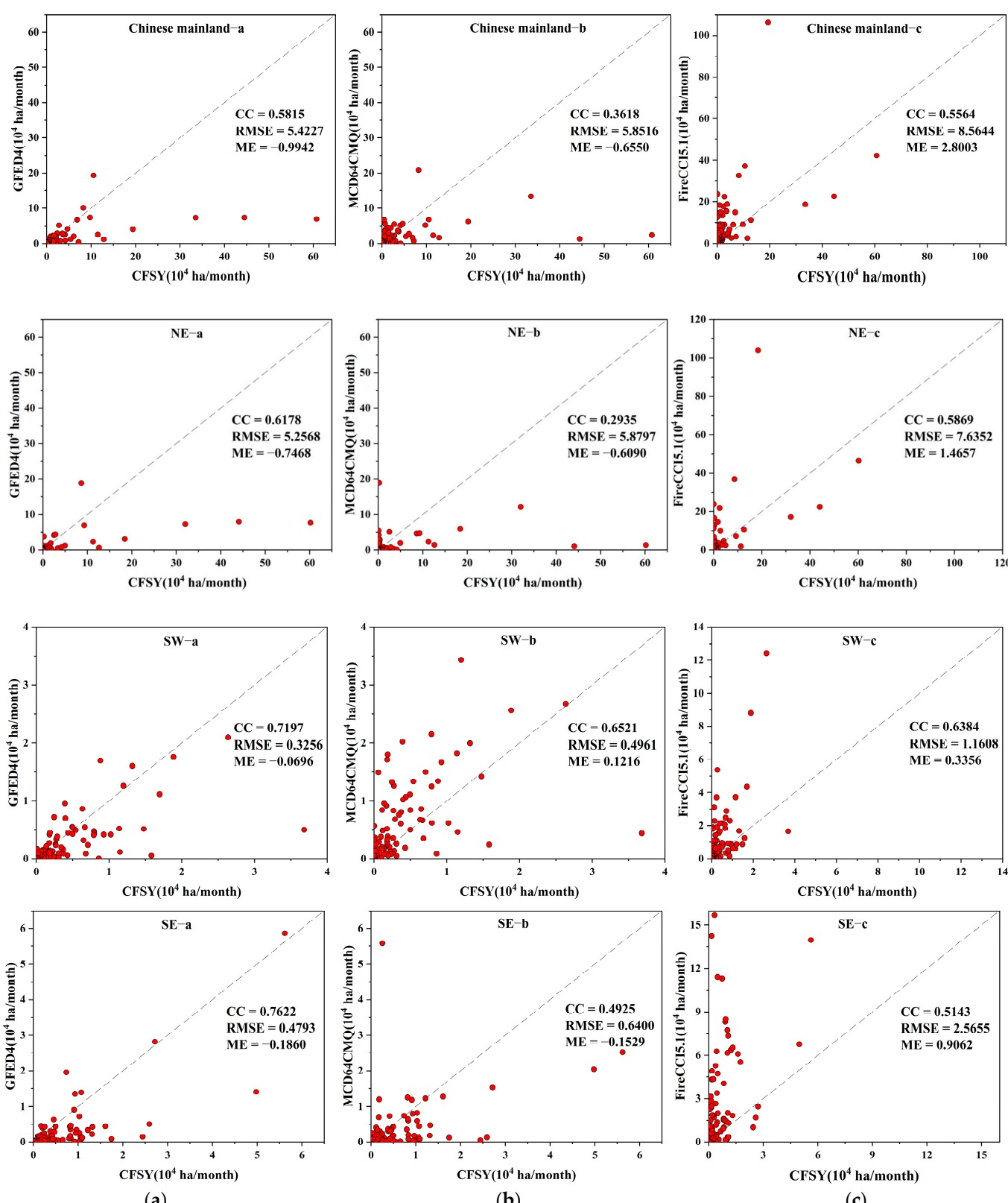

**Figure 6.** Scatter plots of monthly BA in Chinese mainland and the three subregions (NE, SW, and SE) for the CFSY versus the three satellite estimates: (**a**) GFED4, (**b**) MCD64CMQ, and (**c**) FireCCI5.1.

**Table 3.** Statistics of seasonal BA in the three regions from 2001 to 2016 *.

| Season | Region | Index | GFED4 | MCD64CMQ | FireCCI5.1 | | GFED4 | MCD64CMQ | FireCCI5.1 |
|---|---|---|---|---|---|---|---|---|---|
| Spring | NE | CC | **0.9485** | 0.5834 | 0.9401 | Summer | **0.8013** | 0.3895 | 0.6273 |
| | | RMSE | 14.2537 | 23.5476 | **10.0587** | | **0.9782** | 1.2819 | 1.5334 |
| | | ME | −4.8476 | −5.5048 | 4.9139 | | **−0.4208** | −0.6530 | 0.4553 |
| | SW | CC | 0.1492 | 0.0491 | **0.2884** | | **0.8779** | 0.4055 | 0.6813 |
| | | RMSE | **1.4222** | 1.9062 | 2.4264 | | **0.0655** | 0.0872 | 0.0706 |
| | | ME | −0.5817 | 0.7407 | 0.8186 | | −0.0331 | −0.0240 | **−0.0233** |
| | SE | CC | **0.7605** | 0.7459 | 0.4879 | | 0.5234 | **0.6508** | 0.6242 |
| | | RMSE | **1.5739** | 1.8596 | 5.1722 | | 0.2501 | 0.2595 | **0.2020** |
| | | ME | −1.1869 | −1.4365 | 2.3617 | | −0.1222 | −0.1286 | **−0.0666** |
| Autumn | NE | CC | 0.7276 | **0.8112** | 0.7386 | Winter | 0.0900 | 0.0552 | **0.2485** |
| | | RMSE | 5.4413 | **4.4297** | 24.1517 | | **0.1277** | 0.2037 | 0.2929 |
| | | ME | −2.2671 | **−1.2471** | 12.0213 | | **0.0576** | 0.0969 | 0.1980 |
| | SW | CC | 0.4013 | **0.7301** | 0.5146 | | 0.8608 | 0.7150 | **0.8815** |
| | | RMSE | **0.0380** | 0.0676 | 0.2283 | | **0.4163** | 1.0729 | 4.6175 |
| | | ME | −0.0217 | 0.0268 | 0.0869 | | −0.2097 | 0.7151 | 3.1446 |
| | SE | CC | 0.4032 | 0.4070 | **0.4630** | | **0.7689** | 0.5108 | 0.6266 |
| | | RMSE | 0.6833 | **0.6482** | 3.5014 | | **1.2596** | 1.6882 | 7.9389 |
| | | ME | **−0.2510** | −0.2605 | 2.2037 | | −0.7113 | **−0.0090** | 6.3756 |

* The bold represents the best performance, and the units of RMSE and ME are $10^4$ ha/month.

### 3.3. Comparisons of Provincial and Reginal BA

Figure 7 shows the provincial indicator values of the three products. The darker the color for a province is, the better the indicator value was. For CCs, GFED4 performed better than MCD64CMQ and FireCCI5.1 for the SW and SE regions, mainly referring to the Sichuan (Sc) province in the SW region and the provinces of Zhejiang (Zj), Jiangxi (Jx), Hunan (Hun), and Fujian (Fj) in the SE region. GFED4 in the NE region had a high CC (although slightly lower than that of FireCCI5.1), mainly referring to the provinces of Inner Mongolia (IM), Jilin (Jl), and Liaoning (Ln). For RMSEs, GFED4 had lower values than MCD64CMQ and FireCCI5.1 for all the provinces in the three regions. The best performances for GFED4 occurred in the provinces of Tibet, Ln, Sc, and Hainan (Hn). For MEs, FireCCI5.1 had the worst performance. GFED4 had the same performance rank as MCD64CMQ in all the provinces of the NE region and in five of nine provinces for the SW and NE regions. For the remaining provinces of Ln, IM, Sc, and Cq, the performance of GFED4 was slightly lower than that of MCD64CMQ.

Overall, GFED4 had the best performance in RMSEs for all the provinces of the three regions, CCs for the provinces of the SW and SE regions, and MEs for the provinces of the SE region. For MEs, the number of provinces where GFED4 performance was slightly worse than MCD64CMQ was less than two. Therefore, it was concluded that GFEF4 showed better performance at the provincial and regional scales.

### 3.4. Comparison of the Number of Months for Different BA Classes

Referring to the forest fire grades stipulated in the Regulations of the People's Republic of China on Forest Fire Prevention, the BA was divided into four levels: (a) no-fire BA (0 ha/month), (b) common forest fire BA (<100 ha/month), (c) severe forest fire BA (100–1000 ha/month), and (d) disaster forest fire BA (>1000 ha/month). Then, the numbers of months for different BA levels from 2001 to 2016 were counted, and the results are shown in Figure 8.

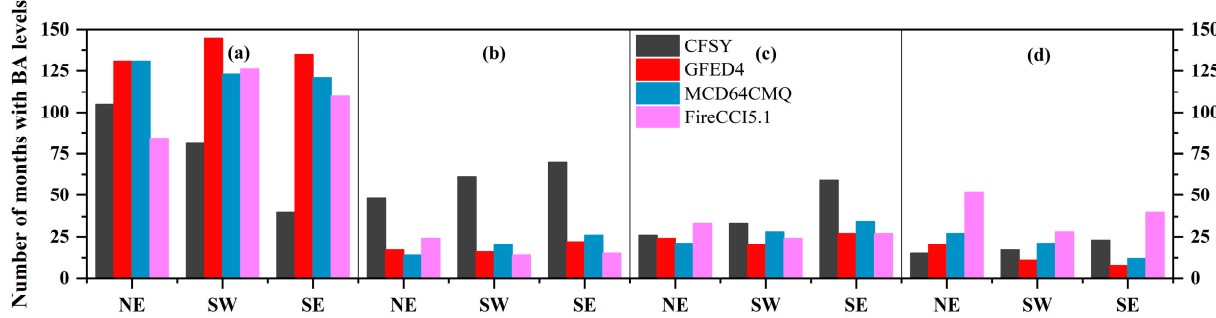

**Figure 7.** Spatial patterns of the provincial indicator values for the three satellite products.

**Figure 8.** Number of months for different BA levels: (**a**) number of no- fire months (0 ha/month), (**b**) number of common forest fire months (<100 ha/month), (**c**) number of severe forest fire months (100–1000 ha/month), and (**d**) number of disaster forest fire months (>1000 ha/month).

Two findings were obtained from the CFSY data in Figure 8. Firstly, the number of months with no fires and common fires was far higher than the numbers of months with severe and disaster fires. Secondly, the SE region had the highest number of months with fires. In terms of satellite product comparisons, all three products detected more no-fire months, which may have been related to the relatively rough spatial scale for satellite data, thereby limiting detection to the minimum fire area. Additionally, GFED4 had a

more consistent number of months with the CFSY than the other products for fires in the NE region. FireCCI5.1 significantly overestimated the disaster fire BA (>1000 ha/month). Overall, all three products had stronger detection abilities for severe and disaster fires than for common fires.

The differences among the three products were also analyzed at the provincial scale. Figure 9 shows the comparisons of the number of months in 16 provinces of the three regions for different levels of BA. For no-fire months (Figure 9a), the underestimation of FireCCI5.1 in the NE region mainly occurred in the Hlj and IM provinces. The overestimates of the three products in the SW were mainly distributed across the Cq, Gz, and Sc provinces, and the overestimates in the SE occurred in all provinces. For the common fire BA (Figure 9b), significant underestimates were mainly distributed in the Jl and Ln provinces in the NE region, Cq province in the SW region, and Hn province in the SE region. Certainly, the three products presented general underestimates in all the provinces of the three regions.

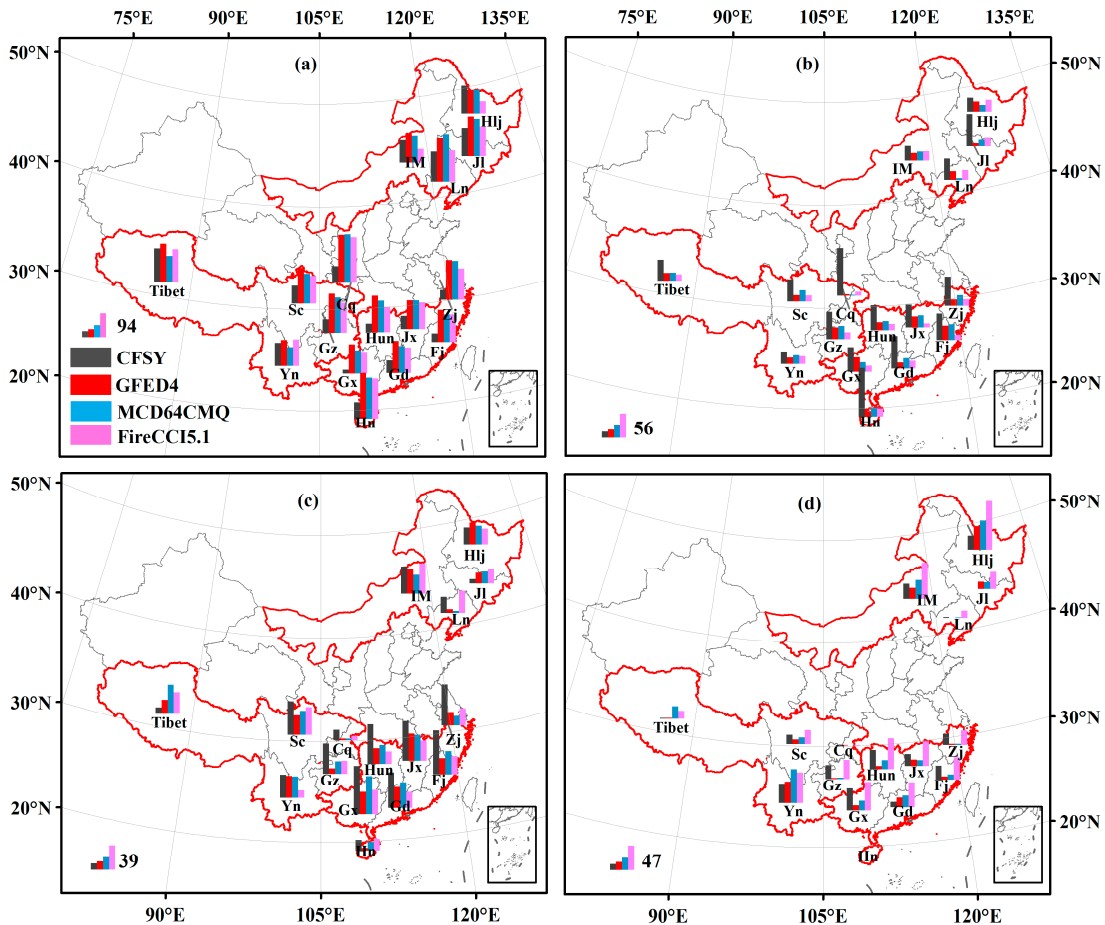

**Figure 9.** Spatial pattern of numbers of months with different levels of BA in 16 provinces of the three regions: (**a**) number of no-fire months (0 ha/month), (**b**) number of common forest fire months (<100 ha/month), (**c**) number of severe forest fire months (100–1000 ha/month), and (**d**) number of disaster forest fire months (>1000 ha/month).

The severe fire BAs (Figure 9c) in the CFSY and the three products for each region were almost the same overall (e.g., Figure 8c), but significant differences existed at the province level. For instance, the NE regions presented overestimates in the Jl province and underestimates in the Ln province; the SW region presented overestimates in Tibet and underestimates in the Cq and Gz provinces; all provinces in the SE region presented underestimates, especially in the Zj province. For the disaster fire BAs (Figure 9d), the overestimate of FireCCI5.1 in the NE region mainly occurred in the Hlj and IM provinces. In

the SE region, the performance of FireCCI5.1 in all of the provinces was slightly higher than that of the CFSY except in Gd province. For GFED4 and MCD64CMQ, their underestimates mainly occurred in the Gz and Sc provinces in the SW region and almost all provinces in the SE region except for the Gd province. There were no disaster fires in the Cq and Hn provinces.

### 3.5. Evaluation of the Three Satellite BA Products with Other Ground Observations

Besides the CFSY, we also used other ground BA observation data to evaluate the performances of the three satellite BA products in the Chinese region. Based on the collected ground observations, the study area is located in the Greater Khingan Mountains of the Inner Mongolia province in the NE region. The specific location lies in 119°36′20″– 125°19′50″E, 47°03′40″–53°20′00″N (Figure 10a). The ground BA data (reference data) from 1990 to 2019 came from the Fire Prevention Office of the Inner Mongolia Forestry and Grassland Bureau in China. Among them, 2003 and 2006 were two disaster fire years; therefore, we selected the forest fire BAs in these two years to evaluate the performances of the three satellite BA products.

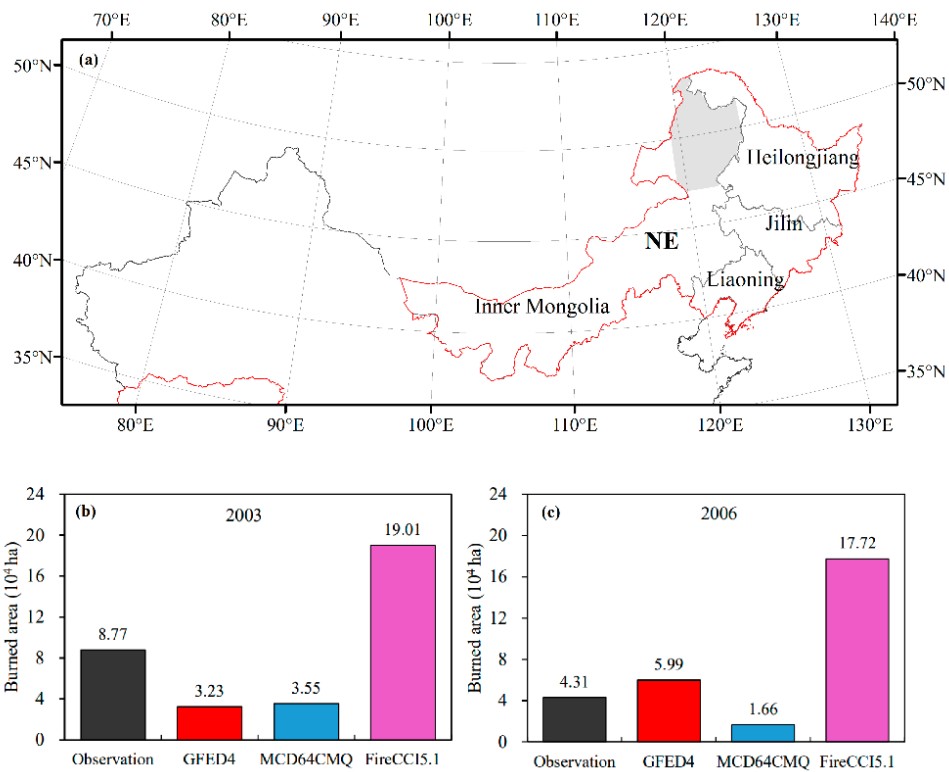

**Figure 10.** Comparisons of three satellite BA products based on a new observation dataset: (**a**) Location of the forest fire region marked with a filled gray block, (**b**) the histograms of the four BA datasets in 2003, and (**c**) the histograms of the four BA datasets in 2006.

The BA comparison results of the three satellite products for 2003 and 2006 are shown in Figure 10b,c. In 2003, the forest BAs detected by GFED4, MCD64CMQ, and FireCCI5.1 were $3.23 \times 10^4$, $3.55 \times 10^4$, and $19.01 \times 10^4$ ha, respectively. Compared with the ground observation of $8.77 \times 10^4$ ha, the FireCCI5.1 estimate was significantly high, and those of GFED4 and MCD64CMQ were closer to the ground observations. In 2006, the forest BAs detected by GFED4, MCD64CMQ, and FireCCI5.1 were $5.99 \times 10^4$, $1.66 \times 10^4$, and $17.72 \times 10^4$ ha, respectively. Compared with the ground observations with $4.31 \times 10^4$ ha, it was clear that GFED4 was the closest to the ground observations. In total, GFED4 performed better than the other two products. The GFED4 estimate for the Chinese region was again proven to be reasonable. It was also demonstrated that the evaluation conclusions of the

three products in the selected Chinese regions, based on the CFSY data, were reasonable and credible.

## 4. Discussion

This study evaluated three satellite BA products in China from 2001 to 2016 based on the reference data derived from the CFSY. The results showed that GFED4 had the best performance on the estimates of BA data followed by MCD64CMQ and FireCCI5.1. MCD61A1, developed for forming GEFD4 and MCD64CMQ, has been proven to have a better performance than other satellite products (e.g., MCD45A1, AVHRR, and FireCCI51) in some regions such as the North American boreal region [19], Indonesia [30], and Greece [25], and even on a global scale [16]. This proves the reasonability of our conclusions. In addition, some studies support our view that there are large differences in the BA values among the three satellite products. For example, Freeborn et al. [21] found that the MODIS and World Fire Atlas (WFA) have distinct differences in the onset, peak, and duration for fires in the Central African Republic, Pessôa et al. [23] proved that the four satellite BA products that they studied presented great divergence in the Amazon, Zhang et al. [28] admitted that MCD64A1 and FireCCI5.1 showed frequent changes in monthly BA peaks for cropland regions in China, and Humber et al. [17] found that the four satellite fire BA products that they studied, including FireCCI5.1 and MCD64A1, varied greatly. Finally, the finding that FireCCI5.1 produced a significant overestimate has also been proven in other studies. For instance, Katagis and Gitas [25] found that FireCCI5.1 overestimated the actual burned area by about 8% in the Mediterranean region. Overall, similar findings in other studies demonstrate that some of our conclusions are reasonable.

Two significantly high values in the CFSY occurred in 2003 and 2006 for the Chinese mainland (Figure 3). Further results found that 90% and 87% of the BAs for these two years were from the NE region, mainly in the spring season in the Hlj province, which could be easily detected by the three satellite products. These fires occurred in a vast expanse of virgin forest in the Hlj province. In addition, it was found from Figures 8 and 9 that the three satellite products significantly underestimated common fire BA (<100 ha) compared with the CFSY. This may be related to the abilities of satellite detection. The spatial scale limitations of satellite data make it impossible to detect the area beneath 20 ha fire [25]. Finally, it was found from Figures 7–9 that the apparent overestimation of FireCCI5.1 in the SE region was a prominent phenomenon, especially for disaster fire BA. This may be related to the albedo changes caused by deforestation rather than fires. The SE region is an area with high forest fragmentation and a high population, and the increased secondary forests and agricultural activities lead to an obvious albedo change; FireCCI5.1 mistook these for fires [31].

The spatial resolution is one of the important factors that affect the detection accuracy of satellite sensors to fire BA. Many studies have found that high spatial resolution could lower the impact of spatial heterogeneity in fire BA and allow to make easier comparisons with ground-based data [22,32,33]. Giglio et al. [34] found that despite the continuous adjustments to the DB mapping algorithm, MCD64A1 still only has a minimum detectable burn size of ~40 ha in cropland, which far exceeds the size of many agricultural water burns. Hall et al. [35] thought that compared to the pixel size of moderate and coarse resolution sensors, small fires only cover a fraction of the pixel area and their optical burn signatures are not sufficiently distinct to be mapped by coarser resolution imagery. Chuvieco et al. [36] pointed out that medium-to-high resolution sensors are required for better understanding global fire impacts and trends, particularly in continents where the proportions of small fire patches are significant. Therefore, it is suggested that the missing small fires (or small BA data) should be recovered through more active fire observations to compensate for the omission errors of the coarse resolution sensors.

Besides the resolution, the detection capabilities of satellite remote sensing products are also affected by different terrain, vegetation, and meteorological conditions. Fornacca et al. [37] found that if there is no proper topographic adjustment of raw satellite images, the

surface area of a pixel will be significantly lower than the real area in a rugged terrain and on steep slopes, further affecting satellite BA accuracy. Vegetation coverage is a dominant factor for forest fires. Generally, we think that higher vegetation coverage means significant fire occurrence rate. However, Ma et al. [38] found that the occurrence probability of forest fires is at a higher value for vegetation coverage between 0.8 and 0.97, while it drops rapidly to the minimum value when the coverage is approximately 0.98. A possible reason for this is that the canopy occlusion in high-coverage forest impedes the detection accuracies of satellite sensors like MODIS. Vegetation type is another factor affecting satellite detection accuracy. Several fires occurring in herbaceous vegetation present the characteristics of burning fast and also recovering fast without spectrally discernable scars. Obviously, these characteristics are not favorable to BA algorithms based on spectral reflectance. Some cloud characteristics such as area, layer thickness, and shift also impact the detection capabilities of satellites. In addition, some human activities such as deforestation, urbanization, and farmland planting change the surface albedo in short periods [31], which may influence the detection capabilities of satellites, especial for sensors based on spectral reflectance.

The comparison between the satellite products with only 0.25-degree spatial resolutions and the official provincial statistics was conducted partly due to the limitations of the spatial scale of official data. More BA product assessment with different spatial resolutions could be pursued in the future. Certainly, more evaluation studies in different locations and seasons will strengthen the comparison conclusions [39]. Additionally, different complex scenarios involving natural conditions and social activities will create more challenges for small fire monitoring, and so, it is necessary to improve algorithms to increase the accuracy of small fire monitoring.

## 5. Conclusions

The suddenness of forest fires in space and time and the difficulty in human monitoring have made satellite products that can provide continuous, large-scale, and timely coverage an increasingly popular choice. Many evaluation studies on satellite BA products have been conducted in the past, but few have focused on the assessment studies for China at relatively long time scales. This study evaluated the accuracies of three publicly available BA grid satellite products (GFED4, MCD64CMQ, and FireCCI5.1) in the main Chinese forest regions from 2001 to 2016 based on the CFSY's official statistics. The main conclusions include the following:

(1) Overall, GFED4 detected the least BA and FireCCI5.1 detected the most BA among the three products. A significantly declining BA trend was demonstrated in the Chinese mainland and in the NE and SE regions at an annual scale. It was also found that the BA in China was mainly dominated by forest fires in the NE region.

(2) Compared with CFSY, GFED4 had the best annual performance in the Chinese mainland and the three regions except for the fact that MCD64CMQ slightly outperformed GFED4 in the SE region. At the monthly scale, GFED4 had also the best performance with the lowest RMSE (except for the SE region, where its performance was slightly worse than that of MCD64CMQ). In contrast, FireCCI5.1 had the worst RMSE and ME in the three regions. Similarly, GFFD4 had the best performance in spring (MAM) and winter (DJF). In summer, there are few BAs and, therefore, the differences among the three products were insignificant. Therefore, it could be concluded that GFED4 performs best in three out of four seasons. Overall, the GFED4 estimate was optimal among those of the three satellite products at the monthly and seasonal scales.

(3) At the provincial and regional scale, GFED4 had the best performance in terms of RMSE for all provinces of the three regions, in CCs for the provinces of the SW and SE regions, and in MEs for the provinces of the SE region. For MEs in the SW and NE regions, the province number of each region was no more than two, and here, GFED4 performance was slightly worse than that of MCD64CMQ. Therefore, it is concluded that GFED4 had a better performance at the provincial and regional scales.

(4)    In terms of the number of months for the four levels of BA, the satellite products perform better for larger BAs (>100 ha). Specifically, it was found that the combined number of months with no fires and common fires was far higher than the combined number of months with severe fires and disaster fires. All three products had stronger detection abilities for severe and disaster fires than for common fires. Regionally, the number of months detected with GFED4 was more consistent with the CFSY than those of the other products for fires in the NE region, and the overestimate of FireCCI5.1 mainly occurred in the disaster fires for the three regions. Additionally, for the severe fire BAs, the results were the same overall between the CFSY and the three products for each region, but significant differences existed at the province level. For the disaster fire BA, the overestimate of FireCCI5.1 in the NE region mainly occurred in the Hlj and IM provinces. The underestimates of GFED4 and MCD64CMQ mainly occurred in the Gz and Sc provinces in the SW region and almost all the provinces in the SE region except for the Gd province. There were no disaster fires in the Cq and Hn provinces.

In addition, the conclusion that GFED4 had the best performance in the China region was also proved using another validated BA dataset. This shows the reasonability of our conclusions. Even advanced monitoring systems and the latest satellite band inversion algorithms have difficulties in accurately quantifying forest fires due to their multiple complexities and various uncontrollable influencing factors. However, this study helps us in understanding the feasibility of satellite BA products in China and promoting the fusion and development of multi-source satellite data in the future.

**Author Contributions:** Conceptualization, X.W. and Z.D.; methodology, X.W.; software, X.W.; validation, X.W., Z.D. and J.L.; formal analysis, Z.D.; investigation, J.L.; resources, X.W.; data curation, X.W.; writing—original draft preparation, X.W.; writing—review and editing, Z.D. and J.L.; visualization, X.W.; supervision, Z.D.; project administration, Z.D. and J.L.; funding acquisition, Z.D. All authors have read and agreed to the published version of the manuscript.

**Funding:** This research was supported by the State Key Laboratory of Earth Surface Processes and Resource Ecology (Grant No. 2022-TS-01), the National Natural Science Foundation of China (Grant No. 42275021), and the Natural Science Foundation of Hunan Province (Grant No. 2023JJ30484).

**Data Availability Statement:** The burned area datasets used in our work can be freely accessed from the follow websites: GFED4, MCD64CMQ: sftp://fuoco.geog.umd.edu/ (accessed on 1 May 2022); FireCCI5.1: https://cds.climate.copernicus.eu/ (accessed on 1 May 2022).

**Conflicts of Interest:** The authors declare no conflict of interest.

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
