# Peer review of "Evaluating the Abilities of Satellite-Derived Burned Area Products to Detect Forest Burning in China"

_remotesensing, doi:10.3390/rs15133260_

Round 1

Reviewer 1 Report

Article: Evaluating the Ability of Satellite-derived Burned Area Products to Detect Forest Burning in China

This paper evaluated the performances of three burned area satellite products of GFED4, MCD64CMQ, and FireCCI5.1, in detecting Chinese forest fire burning from 2001 to 2016 using the monthly official statistics from the China Forestry Statistical Yearbook (CFSY). Such a comprehensive evaluation on different time scales (yearly, monthly, and seasonally) and spatial scales (region and province) can provide important information for researchers to understand the characteristics of these products in the Chinese mainland and support the future development of multi-source satellite data fusion. It is suitable for publication in Remote Sensing. I have only a few minor comments below.

1.     One of the innovations of this study is to assess burned area satellite products (namely, GFED4, MCD64CMQ, and FireCCI5.1) at a relatively long-time scale. Although Line 65-85 introduces cases of burned area satellite data evaluation and Section 2 introduces these three datasets, the relevant conclusions of these researches are not emphasized to further explain why these three datasets were chosen.

2.     Line 128: “and the Giglio et al. [26] MODIS DB”, do you mean “and the MODIS DB [26]”.

3.     Line 169 and 172: “mean error (MAE)” should be changed to “mean error (ME)”.

4.     Line 183: the middle subgraph in Figure 2 missed the Chinese nine-dash line.

5.     Line 217: “Figure 4. Scatter density plots”, do you mean “Figure 4. Scatter plots” as Figure 6 shown?

Author Response

Revision notes for “Evaluating the Ability of Satellite-derived Burned Area Products to Detect Forest Burning in China”

We want to express my sincere thanks for the editors and reviewers on the constructive feedback. The valuable and thoughtful comments from you have certainly helped to improve the presentation and quality of our manuscript. We have updated our paper according to your comments and the detailed responses to the comments are described as follows.

To Review#1
Responses to comments one by one:

Comments: This paper evaluated the performances of three burned area satellite products of GFED4, MCD64CMQ, and FireCCI5.1, in detecting Chinese forest fire burning from 2001 to 2016 using the monthly official statistics from the China Forestry Statistical Yearbook (CFSY). Such a comprehensive evaluation on different time scales (yearly, monthly, and seasonally) and spatial scales (region and province) can provide important information for researchers to understand the characteristics of these products in the Chinese mainland and support the future development of multi-source satellite data fusion. It is suitable for publication in Remote Sensing. I have only a few minor comments below.

  1. One of the innovations of this study is to assess burned area satellite products (namely, GFED4, MCD64CMQ, and FireCCI5.1) at a relatively long-time scale. Although Line 65-85 introduces cases of burned area satellite data evaluation and Section 2 introduces these three datasets, the relevant conclusions of these researches are not emphasized to further explain why these three datasets were chosen.

Response:

We thank the reviewer for pointing this out. We have supplemented some descriptions in the text to address your concerns and hope that it is now clearer. The detailed revision are as follows (See P5, L177-185 in the revised manuscript):

“In summary, MODIS sensor on the Terra and Aqua platforms is one of the important detect tools to global BA data. The MCD64 and FireCCI51 products are often compared in pairs and provide BA data in numerous fire assessment researches. Similarly, GFED4 is widely used to helps the large-scale atmospheric and biogeochemical models to explaining the impacts of the climate changes and land management on fire activities. China has vast territory and diverse climate, and there are few studies on the comprehensive evaluations of the satellite BA products in China rather than its certain regions. Therefore, considering the heterogeneity characteristics of China, these three BA products with long time series were selected to evaluate their detection abilities to the forest fire BA in China.”

  1. Line 128: “and the Giglio et al. [26] MODIS DB”, do you mean “and the MODIS DB [26]”.

Response:

Your expression is corrected. So, we revised this sentence as follows (See P4, L150-154 in the revised manuscript):

“Giglio et al. [13] applied the MODIS DB burned-area mapping algorithm on this dataset to obtain BA maps, and extended the time series back to 1997 by combined it with active fire data from the Tropical Rainfall Measuring Mission (TRMM) Visible and Infrared Scanner (VIRS) and the Along-Track Scanning Radiometer (ATSR) sensors.”

  1. Line 169 and 172: “mean error (MAE)” should be changed to “mean error (ME)”.

Response:

We have fixed the error. ME is the abbreviation of “mean error”, and MAE is the abbreviation of “mean absolute error”. Therefore, your revision suggestion is correct, and we have changed all of the “MAE” to “ME” in the revised manuscript, see P5, L197 and 202 in the revised manuscript.  

  1. Line 183: the middle subgraph in Figure 2 missed the Chinese nine-dash line.

Response:

We have fixed the error. The new Figure 2 has been replaced, see P6, L214 in the revised manuscript.

Figure 2. Spatial patterns of the total BA for the three satellite products in the Chinese mainland.

  1. Line 217: “Figure 4. Scatter density plots”, do you mean “Figure 4. Scatter plots” as Figure 6 shown?

Response:

As your suggestion, we have changed “Scatter density plots” to “Scatter plots” in the revised manuscript, see P8, L251. Additionally, there is no scatter density plots shown in the manuscript.

Reviewer 2 Report

This study evaluated the performance of three publicly available burned area products (GFED4, MCD64CMQ, FireCCI5.1) in detecting forest fires in China during 2001-2016. It used different temporal scales (annual, monthly, seasonal) and spatial scales (regional and provincial) to compare these products with the official monthly statistics from the China Forestry Statistical Yearbook (CFSY). It found that GFED4 was the best burned area product, followed by MCD64CMQ and FireCCI5.1. These results provide important information for users to understand the characteristics of these products in China and support the development of future multi-source satellite data fusion.

Here are some suggestions for improvement:

(1) Due to the limitation of spatial resolution, some small-area fires may be ignored or misjudged, which may result in remote sensing products underestimating fire area, especially in forest-dense or complex terrain areas. It is suggested to discuss in detail the impact of spatial resolution on the research conclusions.

(2) The study only used the China Forestry Statistical Yearbook as reference data, without using other available fire data sources, such as ground observation, high-resolution imagery, drone monitoring, etc., to verify the accuracy of satellite data products. This may have some bias and inaccuracy. Interpretation combined with existing observation data or higher resolution fire spatial data to supplement verification of remote sensing products.

(3) The study did not analyze the detection capability of satellite remote sensing products under different terrain, vegetation and meteorological conditions, which may neglect some influencing factors. Due to not considering the impact of different environmental factors, remote sensing products may perform poorly in some regions or seasons, which may lead to insufficient understanding of the spatiotemporal distribution and change patterns of fire. It is suggested to make appropriate supplements.

Minor editing of English language required

Author Response

Revision notes for “Evaluating the Ability of Satellite-derived Burned Area Products to Detect Forest Burning in China”

We want to express my sincere thanks for the editors and reviewers on the constructive feedback. The valuable and thoughtful comments from you have certainly helped to improve the presentation and quality of our manuscript. We have updated our paper according to your comments and the detailed responses to the comments are described as follows.

To Review#2
Responses to comments one by one:

Comments: This study evaluated the performance of three publicly available burned area products (GFED4, MCD64CMQ, FireCCI5.1) in detecting forest fires in China during 2001-2016. It used different temporal scales (annual, monthly, seasonal) and spatial scales (regional and provincial) to compare these products with the official monthly statistics from the China Forestry Statistical Yearbook (CFSY). It found that GFED4 was the best burned area product, followed by MCD64CMQ and FireCCI5.1. These results provide important information for users to understand the characteristics of these products in China and support the development of future multi-source satellite data fusion.

Here are some suggestions for improvement:

  1. Due to the limitation of spatial resolution, some small-area fires may be ignored or misjudged, which may result in remote sensing products underestimating fire area, especially in forest-dense or complex terrain areas. It is suggested to discuss in detail the impact of spatial resolution on the research conclusions.

Response:

We thank the reviewer for pointing this out. We have revised the text to address your concerns and hope that it is now clearer. Please see P15, L427-440 in the revised manuscript. The details are as follows:

“The spatial resolution is one of important factors that affect the detection accuracy of satellite sensor to fire BA. Many studies have found that the high spatial resolution could lower the impact of spatial heterogeneity in fire BA, and make easier comparisons with ground-based data [22,33,34]. Giglio et al. [35] found that despite the continuous adjustment to the DB mapping algorithm, but MCD64A1 still only have a minimum detectable burn size of ~40 ha in cropland, which far exceeds the size of many agricultural water burns. Hall et al [36] thought that compared to the pixel size of moderate and coarse resolution sensors, the small fires only cover a fraction of the pixel area and their optical burn signatures are not sufficiently distinct to be mapped by the coarser resolution imagery. Chuvieco et al. [37] pointed out that the medium to high resolution sensors are required for better understanding the global fire impacts and trends, particularly in continents where the proportion of small fire patches is significant. Therefore, it is suggested that the missing small fires (or small BA data) should be recovered through more active fire observations to compensate for the omission errors of the coarse resolution sensors.”

  1. Van der Werf, G. R.; Randerson, J. T.; Giglio, L. et al. Global fire emissions estimates during 1997–2016. Earth Syst. Sci. Data. 2017, 9, 697-720. https://doi.org/10.5194/essd-9-697-2017
  2. Eibedingil, I.G.; Gill, T.E.; Van Pelt, R.S.; Tong, D.Q. Comparison of Aerosol Optical Depth from MODIS Product Collection 6.1 and AERONET in the Western United States. Remote Sens.202113, 2316. https://doi.org/10.3390/rs13122316
  3. Giglio, L.; Randerson, J. T.; Van Der Werf, G. R. Analysis of daily, monthly, and annual burned area using the fourth-generation global fire emissions database (GFED4). J. Geophys. Res-Biogeo. 2013, 118, 317-328. https://doi.org/10.1002/jgrg.20042.
  4. Hall, J. V.; Argueta, F.; Giglio, L. Validation of MCD64A1 and FireCCI51 cropland burned area mapping in Ukraine. Int. J. Appl. Earth Obs. & Geoinf. 2021, 102, 102443. https://doi.org/10.1016/j.jag.2021.102443.
  5. Chuvieco, E.; Roteta, E.; Sali, M. et al. Building a small fire database for Sub-Saharan Africa from Sentinel-2 high-resolution images. Sci. Total Environ. 2022, 845, 157139. http://dx.doi.org/10.1016/j.scitotenv.2022.157139.

  1. The study only used the China Forestry Statistical Yearbook as reference data, without using other available fire data sources, such as ground observation, high-resolution imagery, drone monitoring, etc., to verify the accuracy of satellite data products. This may have some bias and inaccuracy. Interpretation combined with existing observation data or higher resolution fire spatial data to supplement verification of remote sensing products.

Response:

It’s really a good suggestion. To supplement the verification of three BA products in our study, we chose a relatively small region (belongs to the Greater Khingan Mountains of Inner Mongolia province) in NE of China, where some new ground observations are available. Based on the observations, the BA data of two disaster fire years (2003 and 2006) were selected to verify the performance of the satellite fire BA products. The comparison results demonstrated that the GFED4 still performed better than the other two products, which proved that the evaluation conclusions for three satellite BA products on China based on the CFSY data are reasonable and credible. The specific supplement was added in the revised manuscript, see P14, L369-392.

3.5. Evaluation of the three satellite BA products with other ground observations

Besides the CFSY, we also used other ground BA observation data to evaluate the performances of three satellite BA products in the Chinese region. Based on the collected ground observations, the study area is located in the Greater Khingan Mountains of Inner Mongolia province in the NE region. The specific location lies in 119°36′20"-125°19′50"E, 47°03′40"-53°20′00"N (Fig.10a). The ground BA data (reference data) from 1990 to 2019 came from the Fire Prevention Office of the Inner Mongolia Forestry and Grassland Bureau in China. Among them, 2003 and 2006 were are two disaster fire years; therefore, we selected the forest fire BAs in these two years to evaluate the performances of the three satellite BA products.

The BA comparison results of the three satellite products for 2003 and 2006 are shown in Figure 10b-c. In 2003, the forest BAs detected by GFED4, MCD64CMQ, and FireCCI5.1 were 3.23×104, 3.55×104, and 19.01×104 ha, respectively. Compared with the ground observation of 8.77×104 ha, the FireCCI5.1 estimate was significantly high, and those of GFED4 and MCD64CMQ were closer to the ground observations. In 2006, the forest BAs detected by GFED4, MCD64CMQ, and FireCCI5.1 were 5.99×104, 1.66×104, and 17.72×104 ha, respectively. Compared with the ground observations with 4.31×104 ha, the GFED4 was the closet to the ground observations. In total, GFED4 performed better than the other two products. The GFED4 estimate for the Chinese region was again proved to be reasonable. It was also demonstrated that the evaluation conclusions of three products on Chinese region based on the CFSY data are reasonable and credible. 

Fig.10 Location of forest fire region with filled gray block and the histograms of the four BA datasets in 2003 and 2006.”

  1. The study did not analyze the detection capability of satellite remote sensing products under different terrain, vegetation and meteorological conditions, which may neglect some influencing factors. Due to not considering the impact of different environmental factors, remote sensing products may perform poorly in some regions or seasons, which may lead to insufficient understanding of the spatiotemporal distribution and change patterns of fire. It is suggested to make appropriate supplements.

Response:

We have supplemented the discussion related to the influencing factors affecting the detection capability of satellite remote sensing products. The specific revision has been added the revised manuscript, See P16, L441-459.

“Besides the resolution, the detection capability of satellite remote sensing products is also affected by the different terrain, vegetation and meteorological conditions. Fornacca et al. [38] found that if there is not a proper topographic adjustment of the raw satellite images, the surface area of a pixel will significantly lower than the real area in a rugged terrain and on steep slopes, further affecting the satellite BA accuracy. The vegetation coverage is a dominated factor for forest fires. Generally, we think that higher vegetation coverage have significant fire occurrence rate. However, Ma et al. [39] found that the occurrence probability of forest fires is at a higher value for the vegetation coverage between 0.8 and 0.97, while it drops rapidly to the minimum value when the coverage is approximately 0.98. The possible reason is that the canopy occlusion in high-coverage forest impedes detection accuracies of satellite sensors like MODIS. The vegetation type is another factor affecting the satellite detection accuracy. Several fires occurred in herbaceous vegetation present the characteristics that burns fast and recover also fast without spectrally-discernable scars. Obviously, it is not favorable to the BA algorithms based on spectral reflectance. Some cloud characteristics such as area, the layer thickness, and shift also impact the detection capability of satellite. In addition, some human activities such as deforestation, urbanization, and farmland planting change the surface albedo in short period [32], which may influence the detection capability of satellite, especial for the sensors based on spectral reflectance. 

  1. Fornacca, D.; Ren, G.; Xiao, W. Performance of three MODIS fire products (MCD45A1, MCD64A1, MCD14ML), and ESA Fire_CCI in a mountainous area of northwest Yunnan, China, characterized by frequent small fires. Remote Sens.20179, 1131. https://doi.org/10.3390/rs9111131
  2. Ma, W.; Feng, Z.; Cheng, Z.; Chen, S.; Wang, F. Identifying forest fire driving factors and related impacts in China using random forest algorithm. Forests202011, 507. https://doi.org/10.3390/f11050507
  3. Comments on the Quality of English Language. Minor editing of English language required

Response:

We have re-checked the text and then submitting the manuscript to the professional English editing company for polishing. Please refer to the attachment for the proof material of polishing.

Reviewer 3 Report

Paper introduces a comparison of different burned area products on temporal and spatial basis. Papers' novelty and significance is limited to comparison of products, no new approach or methodology is introduced. Thus maybe it can be evalauted as a Communication paper. My spesific comments are as follows:

1. Sentence strucuture of the Abstract is is in low quality. First sentence should be completly re-written. 

2. In line 16 how these forest fire areas are defined? Moreover providing directions instead of locational descriptions seemed unusable to me.

3. All point based findings in the abstract should be supported by quantitative metrics nemerically.

4. Lines 44-60  is too long. maybe just mention about the options in a short sentence, and extend  (4) satellite remote sensing based detection with providing different approaches in literature.

5. Last paragraph should focus on waht is new or novel in this study and why findings of this study will be important.

6. I guess using elevation model based view in Figure 1 is not much relevant to the study aim. A thematic map or satellite image view that we can see the forest distribution is better. And again the same suggesion, instead of using directions, please name study regions (maybe A - B- C).

7. The China Forestry Statistical Yearbook (CFSY) data property is not fully given? by which method BA data is determined? what is the accuracy of this detection etc. are important as Authors used these data as reference.

8. Authors should clarify why they selected current statistical metrics among other several metrics. 

9. Lines 254 and 262 are replicating each other and more importantly  how the amount of BA value in one season can determine the perfomance of one method?

10. For table 3, what is the unit for RMSE and ME. and more importantly does these metrics are calculated for cumulative area information of BA for each region and for different seasons, which means Authors only performed an areal comparison. 

The English of the paper is very difficult to understand, there are so many problematic sentences that has structural defects. My advice is a complete check and whole revision of paper for English clarity.

Author Response

Revision notes for “Evaluating the Ability of Satellite-derived Burned Area Products to Detect Forest Burning in China”

We want to express my sincere thanks for the editors and reviewers on the constructive feedback. The valuable and thoughtful comments from you have certainly helped to improve the presentation and quality of our manuscript. We have updated our paper according to your comments and the detailed responses to the comments are described as follows.

To Review#3
Responses to comments one by one:

Comments: Paper introduces a comparison of different burned area products on temporal and spatial basis. Papers' novelty and significance is limited to comparison of products, no new approach or methodology is introduced. Thus maybe it can be evaluated as a Communication paper. My specific comments are as follows:

  1. Sentence structure of the Abstract is in low quality. First sentence should be completely re-written. 

Response:

We apologize for the low quality of the Abstract. We have rewritten the Abstract and text, so that the sentence structures of the Abstract and text have significant improvement.   The first sentence is revised as follows:

“Fire plays a prominent role in the construction and destruction of ecosystems, and accurate estimation of the burned area (BA) after a fire occurrence is of great significance to protect ecosystems and save people’s lives and property.”

  1. In line 16 how these forest fire areas are defined? Moreover, providing directions instead of locational descriptions seemed unusable to me.

Response:

We thank the reviewer for pointing this out. The forest fire areas in this study were quantified by overlapping the forest coverage and the satellite fire BA distribution.

In addition, the three regions of North-east (NE), South-west (SW), and South-east (SE) have been usually used in many studies. For their widely usages, one of the reasons may be that the three regions exist significant difference in precipitation, temperature, and vegetation etc. So, to avoid confusing readers, we have added the new descriptions on the three regions, particularly reflect their differences in climate and vegetation. The specific revisions are as follows (see P3, L104-121 in the revised manuscript).  

“The forest fire areas were quantified by overlapping the forest coverage and the satellite fire BA distribution.”

“The three regions exist significant differences in climate and vegetation. Specifically, the NE region is characterized by higher windspeed, colder climate, heavier snowfall but less rainfall compared to that in Southern China. There are many mountains with an altitude of 300 to 1400 meters in the north of NE, primarily covered by the deciduous and coniferous forests in cold temperature zones. The eastern plains with an altitude below 200 meters are densely populated and primarily covered by the deciduous mixed broadleaf–conifer forests in temperate zone. The NE region has extensive forests that are rarely inhabited by humans, forest fires is characterized by rapid spread and extensive burned area; the SW region is characterized by cloudy and foggy in spring and autumn, with annual precipitation ranged from 600 to 2300 mm. The elevation of most areas is between 500 and 6000 meters, and the vegetation displays a distinct vertical distribution pattern from subtropical broad-leaved evergreen forest to alpine grassland. Furthermore, forests are primarily distributed in high mountain valleys or dry river valleys, which results in the unpredictability and uncontrollability of forest fires due to the complex terrain and climatic background of the region; the SE region mainly consists of hills below 500 meters, characterized by large amounts and high intensity of rainfall over 800 mm. It mainly covered by subtropical broad-leaved evergreen forest, and the high density of forest coverage makes it an important production base for Chinese specialty forest products.”

  1. All point based findings in the abstract should be supported by quantitative metrics numerically.

Response:

We thank the reviewer for pointing this out. We have made some quantitative metric description in the Abstract. For the declining BA trend, we add the description on slopes ranged from -3,821.1 ha/year for MCD64CMQ to -33,218 ha/year for the CFSY; the slopes for the NE region ranged from -3,821.1 ha/year for MCD64CMQ to -33,218 ha/year for the CFSY; and the slopes for the SE region ranged from -594.24 ha/year for GFED4 to -3,162.1 ha/year for the CFSY; For the annual comparisons, the GFED4 with the best performance has a lowest RMSE of 23.9×104 ha/year and the highest CC of 0.83. Similarly, the monthly metrics for GFED4 have been added in the Abstract. The specific revision sees P1, L18-38 in the revised manuscript. The details are as follows:

“The main results are as follows: (1) A significant declining BA trend was demonstrated in the whole study area and in the NE and SE subregions. Specifically, the slopes for the whole area ranged from -3,821.1 ha/year for MCD64CMQ to -33,218 ha/year for the CFSY; the slopes for the NE region ranged from -3,821.1 ha/year for MCD64CMQ to -33,218 ha/year for the CFSY; and the slopes for the SE region ranged from -594.24 ha/year for GFED4 to -3,162.1 ha/year for the CFSY. The BA in China was mainly dominated by forest fires in the NE region, especially in 2003 and 2006 when this region accounting for 90% and 87%, respectively. (2) Compared with the CFSY, GFED4 had the best performance at the yearly scale with an RMSE of 23.9×104 ha/year and CC of 0.83. Similarly, at the monthly scale, GFED4 had also the best performance for the three regions with the lowest RMSE ranging from 0.33×104 to 5.4×104 ha/month, far lower than that of FireCC5.1 which ranged from 1.16×104 to 8.56×104 ha/month (except for the SE region where it was slightly worse than MCD64CMQ). At the seasonal scale, GFFD4 had the best performance in spring and winter. It was also noted that the fewer BAs in summer make it insignificant for the difference among the products. (3) Spatially, GFED4 had the best performance in RMSEs for all provinces of the three regions, in CCs for the provinces of the SW and SE regions, and in MEs for the provinces of the SE region. (4) All three products had a stronger detection ability for severe and disaster fires than for common fires. Additionally, GFED4 had a more consistent number of months with the CFSY than the other products in the NE region. Moreover, the conclusion that GFED4 had the best performance in the China region was also proved using other validated BA datasets. These results will help us to understand the BA detection abilities of the satellite products in China and promote the further development of multi-source satellite fire data fusion.”

  1. Lines 44-60 is too long. maybe just mention about the options in a short sentence, and extend (4) satellite remote sensing based detection with providing different approaches in literature.

Response:

As you suggestion, we have deleted the more detailed descriptions on forest fire detection methods (1)-(3), and extend (4) the description on the development of the satellite remote sensing based detection methods experiencing from the initial the active fire detection, to BA mapping method based on reflectance, brightness temperature and a vegetation index, to the final hybrid algorithms combing mapping method and active fire data.  The specific revision sees P2, L49-70 in the revised manuscript. The details are as follows:

 “There are four common types of forest fire detection methods [4]: (1) ground patrol [5]; (2) near-surface detection [6]; (3) aircraft patrol [7]; and (4) satellite remote sensing detection. Initially, a number of coarse- and medium-resolution sensors were used to detect active fires, such as the Advanced Very High Resolution Radiometer (AVHRR) [8], the Visible and Infrared Scanner (VIRS) [9], and the Moderate Resolution Imaging Spectro-radiometer (MODIS) [10]. However, the active fire products only detected the location and timing of burning fires when the satellite passed over. Obviously, the detected BA data was discontinuous and unreliable. Then, the BA mapping method was developed. For instance, Kasischeke and French [11] generated a BA map for Alaskan boreal forest fires during 1990 to 1991 by applying the differencing method to 15-day AVHRR Normalized Difference Vegetation Index (NDVI) data; and Barbosa et al. [12] mapped BAs in Africa using daily 5 km AVHRR imagery and information on changes in reflectance, brightness, temperature, and a vegetation index. However, these methods do not exploit active fire information and thus create inaccuracies in the BA products. Hybrid algorithms can supplement the drawbacks of previous remote sensing methods, which combines the advantages of remotely sensed indicators (e.g., surface reflectance, surface temperature, NDVI) and active fire maps. Giglio et al. [13] presented an automated method of BA mapping by combining 500 m MODIS imagery and 1 km MODIS active fire observations BA data. Zhang et al. [14] constructed a fire label dataset using the VNP14IMG fire product and Himawari-8 multiband data that includes active fires. Additionally, the products like GFED4, MCD64CMQ, and FireCCI5.1 used in this study were developed using the hybrid algorithm.”

  1. Last paragraph should focus on what is new or novel in this study and why findings of this study will be important.

Response:

We thank the reviewer for pointing this out. We supplemented the descriptions on the defects of previous studies and the novelty of our studies in the last paragraph of the introduction section. We think that there are few studies on the evaluations of the satellite fire BA products for a more comprehensive forest area in China, usually focusing on certain a region. Moreover, the evaluation periods for previous studies were relatively short, usually only three to six years. Therefore, this study evaluates the performances of three popular satellite BA products on the three large forest fire-prone regions of China, covering a relatively long period from 2001 to 2016. The supplementary descriptions see P2, L84-89 in the revised manuscript. The details are as follows:

“There appears to be few studies on the evaluations of the satellite fire BA products for a more comprehensive forest area in China, usually focusing on certain a region. Moreover, the evaluation periods were relatively short, usually only three to six years.

Therefore, this study evaluates the performances of three popular satellite BA products on the three large forest fire-prone regions of China, covering a relatively long period from 2001 to 2016.”

  1. I guess using elevation model based view in Figure 1 is not much relevant to the study aim. A thematic map or satellite image view that we can see the forest distribution is better. And again the same suggestion, instead of using directions, please name study regions (maybe A - B- C).

Response:

We apologize that our original Figure 1 did not show forest areas. We have modified the Figure 1 with land cover and hope that it is now clearer.

Figure 1. Land cover map of China and the three study areas including Northeast (NE) region, Southwest (SW) region, and Southeast (SE) region.

We appreciate the reviewer’s insightful suggestion for the name of study regions. However, we have decided not to modify the name of study regions after our careful consideration. The reasons have been addressed in response to the previous second question.

  1. The China Forestry Statistical Yearbook (CFSY) data property is not fully given? by which method BA data is determined? what is the accuracy of this detection etc. are important as Authors used these data as reference.

Response:

We thank the reviewer for pointing this out. The CFSY was published the National Forestry and Grassland Administration of China (NFGA) every year. It provides the forest fire information mainly includes the number of forest fires, burned area (BA), and corresponding economic losses.

The forest fire BA dataset in the CFSY was collected from the NFGA’s subordinate administrative departments such as the provincial, municipal, and county forestry bureaus.

As a government agency, one of the duties of the NFGA is to record the forest fire information like BA by the specialized officers; therefore, the accuracy of the data issued by them is high.

   The descriptions above have been added into the revised manuscript (Please see P4, L134-141 in the revised manuscript). The details are as follows:

“The China Forestry Statistical Yearbook (CFSY) published by the National Forestry and Grassland Administration of China (NFGA) provides information on forest fires every year. It refers to the number of forest fires, the burned area (BA), and the corresponding economic losses. The forest fire dataset in the CFSY (accessed from www.chinayearbooks.com) was collected from the NFGA’s subordinate administrative departments such as the provincial, municipal, and county forestry bureaus. As a government agency, one of the duties of the NFGA is to record the forest fire information like BA by the specialized officers; therefore, the accuracy of the data issued by them is high.”

  1. Authors should clarify why they selected current statistical metrics among other several metrics. 

Response:

In fact, we have noted the commonly used BA statistical metrics are the commission error ratio, the omission error ratio, and even Dice coefficient combined previous two metrics. However, in this study, we adopted the commonly used metrics (RMSE, ME, and CC) in the field of remote sensing. The reason is related to the scale of reference data from CFSY. The reference data of CFSY presented in the form that each province (irregular boundary) has a value at each month. The province has a large area, usually including hundreds of pixels. It is unreasonable to quantify the difference between a monthly BA value for province and the hundreds of pixels of BA values using the commonly used hitness metrics (the commission error ratio, the omission error ratio, and even Dice coefficient). By contraries, the metrics of RMSE, ME, and CC are more suitable for evaluation. Specifically, we added some explanations about why used the metrics of RMSE, ME, and CC, rather than the metrics of the commission error ratio, the omission error ratio, and the Dice coefficient. The revisions see P5, L198-201 in the section of 2.3.2 “Validation Metrics” in the revised manuscript. The details are as follows:      

“Noted that there are some commonly used BA evaluation metrics such as the commission error ratio, the omission error ratio, and the Dice coefficient combined previous two metrics [15,19,20,25]. However, limited by the scale of referenced dataset, they are not used instead of RMSE, ME, and CC in this study.”

  1. Lines 254 and 262 are replicating each other and more importantly how the amount of BA value in one season can determine the performance of one method?

Response:

We have deleted the sentence that “Noted that the largest BA occurred in spring, ……”. Please see P11, L299 in the revised manuscript.

In addition, what you said is correct, and therefore we revised the last sentence from “Noted that the largest BA occurred in spring, and therefore GFED4 had the best performance on an annual scale” to “Overall, GFED4 had the best performance on the seasonal scale.”

  1. For table 3, what is the unit for RMSE and ME. and more importantly does these metrics are calculated for cumulative area information of BA for each region and for different seasons, which means Authors only performed an areal comparison. 

 Response:

Firstly, we have added the units for RMSE and ME in table 3. The specific revisions see P11, L302 in the revised manuscript.

Secondly, as you understand, these metrics in table 3 were calculated for the cumulative area information of BA for each region and for different seasons. Besides that, the seasonal BAs from 2001 to 2016 were calculated as the metric formulas. 

Comments on the Quality of English Language

  1. The English of the paper is very difficult to understand, there are so many problematic sentences that has structural defects. My advice is a complete check and whole revision of paper for English clarity.

Response:

We have re-checked and re-written the text including sentences and the paragraph structure, and then submitting the manuscript to the professional English editing company for polishing. Please refer to the attachment for the proof material of polishing.

Round 2

Reviewer 2 Report

The paper has been revised according to the opinions of the manuscript review and agreed to be published.

Author Response

Thanks for your constructive comments, and it helps improve the quality of my manuscripts.

Reviewer 3 Report

Authors performed extensive revisions that meets my concerns. Authors provided good explanations about the limitations where they can not improve further.

My only question is that I believe The China yearbook is high accuracy but it should be better to explain how the BA data is collected, (eg filed work or high resolution images) or defining boundary with GPS etc or deriving from image digitization. What is the form of last product etc. 

Author Response

Comments: Authors performed extensive revisions that meets my concerns. Authors provided good explanations about the limitations where they can not improve further.

Response: Thanks for your constructive comments, and it helps improve the quality of my manuscripts a lot. The new response and revision are as follows.

Point 1: My only question is that I believe The China yearbook is high accuracy but it should be better to explain how the BA data is collected, (eg filed work or high resolution images) or defining boundary with GPS etc or deriving from image digitization. What is the form of last product etc.

Response 1: We thank the reviewer for pointing this out. We have revised the text to address your concerns and hope that it is now clearer. Please see lines 140-149. The details are as follows:

“As a government agency, one of the duties of the NFGA is to record the forest fire infor-mation like BA by the specialized officers. The forest BA data are mainly collected through field work combined with some advance technological means. Specifically, with the help of the telescope, unmanned aerial vehicle, helicopter, and even satellite remote sensing, some high resolution fire images were first obtained. Then, the BA boundary and area data were collected from the images with some advanced technologies such as the GPS positioning and the computer image digitization. Finally, the BA data was recorded in the CFSY in the form of digital tables. Noted also that some other fire-related data like fire frequency, casualties, firefighting funds, and firefighting manpower were recorded in the CFSY.”